# A Case Report of Primordial Odontogenic Tumor That Required Distinction from a Dentigerous Cyst

Sawako Ono [1,*], Hotaka Kawai [2], Shintaro Sukegawa [2,3], Kiyofumi Takabatake [2], Keisuke Nakano [2], Hitoshi Nagatsuka [2] and Tadashi Yoshino [4]

1 Department of Pathology, Okayama University Hospital, Okayama 700-0914, Japan
2 Department of Oral Pathology and Medicine, Graduate School of Medicine, Dentistry and Pharmaceutical Sciences, Okayama University, Okayama 700-8558, Japan; de18018@s.okayama-u.ac.jp (H.K.); gouwan19@gmail.com (S.S.); gmd422094@s.okayama-u.ac.jp (K.T.); pir19btp@okayama-u.ac.jp (K.N.); jin@okayama-u.ac.jp (H.N.)
3 Department of Oral and Maxillofacial Surgery, Kagawa Prefectural Central Hospital, Takamatsu 760-0065, Japan
4 Department of Pathology and Medicine, Graduate School of Medicine, Dentistry and Pharmaceutical Sciences, Okayama University, Okayama 700-8558, Japan; yoshino@md.okayama-u.ac.jp
* Correspondence: de19008@s.okayama-u.ac.jp; Tel.: +81-86-235-6651; Fax: +81-86-235-6654

**Abstract:** Primordial odontogenic tumor (POT) is a rare odontogenic tumor characterized by a variably cellular loose fibrous tissue with areas similar to the dental papilla and covered by cuboidal to columnar epithelium. We herein report a case of POT in a 14-year-old boy. Computed tomography (CT) exhibited a round cavity with a defined cortical border circumscribing the tooth of the second molar. However, the gross finding was a solid mass, not a cyst. Histologically, the tumor consisted of dental papillalike myxoid connective tissue covered by columnar epithelium. Therefore, although the clinical diagnosis was dentigerous cyst (DC), we diagnosed POT based on histologic findings. Clinical findings of POT resemble DC, but the clinical behavior of POT is different to DC, such as cortical expansion and root resorption of teeth. Therefore, histological differentiation of POT from DC is critical for accurate diagnosis.

**Keywords:** primordial odontogenic tumor; dentigerous cyst; odontogenic tumor

## 1. Introduction

A primordial odontogenic tumor (POT) is a recently described tumor, which is classified as a benign, mixed odontogenic tumor in the fourth edition of the World Health Organization classification of Head and Neck Tumors in 2017 [1]. Radiologically, POT demonstrates a well-defined jaw lesion with the involved tooth. It has been often clinically diagnosed as a dentigerous cyst (DC) [2]. However, as POT may cause cortical expansion with displacement and root resorption of neighboring teeth [3], distinguishing between DC and POT is important. To date, no reports have described the difference between POT and DC. Therefore, histological and immunohistochemical differences between POT and DC are not clear. Here, we report a case that was clinically diagnosed as DC but was diagnosed to be POT by histological and immunohistochemical findings.

## 2. Case Report

A 14-year-old boy with a history of retarded permanent tooth eruption visited the Department of Oral and Maxillofacial Surgery, Kagawa Prefectural Central Hospital, in February 2018. Computed tomography (CT) exhibited a round cavity with a defined cortical border circumscribing the tooth of the second molar (Figure 1). Thus, we clinically suspected DC. Macroscopically, the lesion was a well-defined, white nodule measuring $10 \times 7 \times 3.5$ mm (Figure 2). Histologically, most of the tumor was composed of cellular loose fibrous connective tissue with a myxoid extracellular matrix and the surface was

covered by columnar epithelium. In the fibrous tissue, spindle or stellate cells were observed. The nuclei of the columnar epithelial cells were arranged close to the basement membrane and the cytoplasm was amphoteric. In some regions, nests of epithelium resembled dental lamina, in which the nuclei of epithelial cells were displaced away from the basement membrane and the cytoplasm were vacuolated (Figures 3 and 4). There were no findings of calcified areas such as dentin formation, and odontogenic epithelial islands or cords.

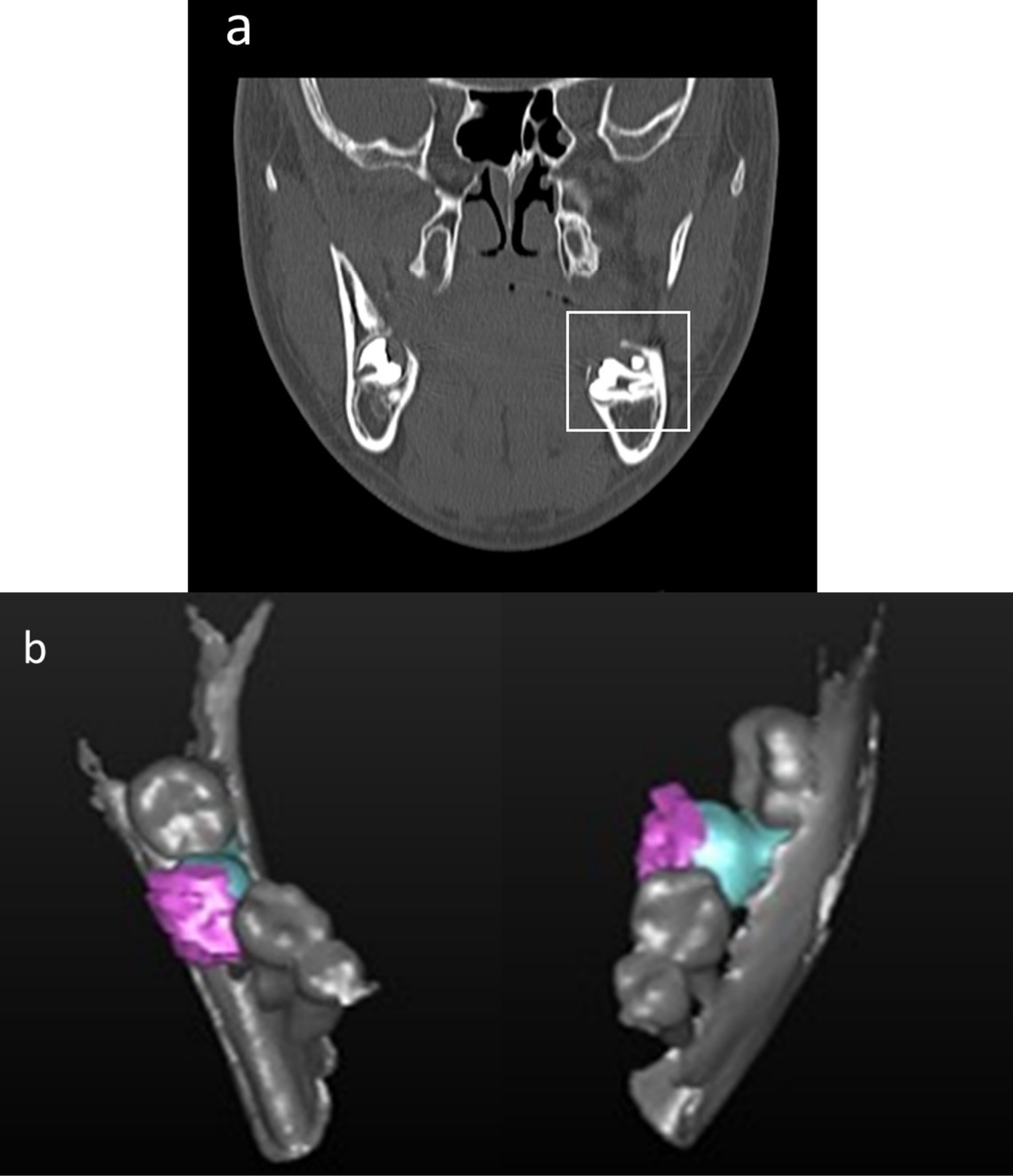

**Figure 1.** Radiographic findings of primordial odontogenic tumor. (**a**) The lesion presents in left side of the posterior mandible as a dentigerous cystlike, well-circumscribed radiolucency associated with an unerupted molar in computed tomography. (**b**) The lesion localizes around the crown of the second molar in 3D format (the lesion: purple; the second molar: blue).

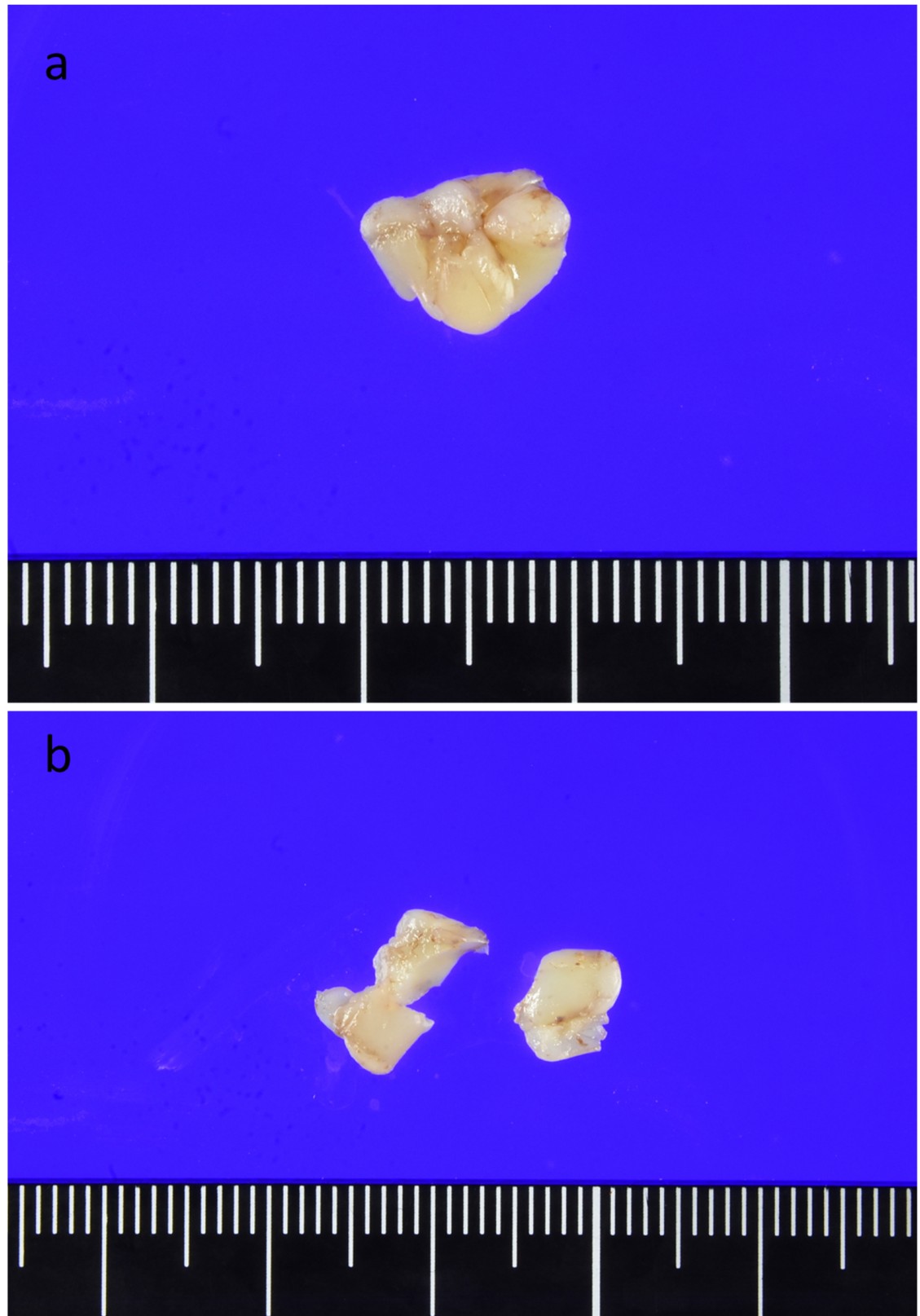

**Figure 2.** Macroscopic findings of primordial odontogenic tumor. (**a**) The tumor was a white mass measuring 10 × 7 × 3.5 mm. (**b**) The cut surface was a solid, not cystic lesion.

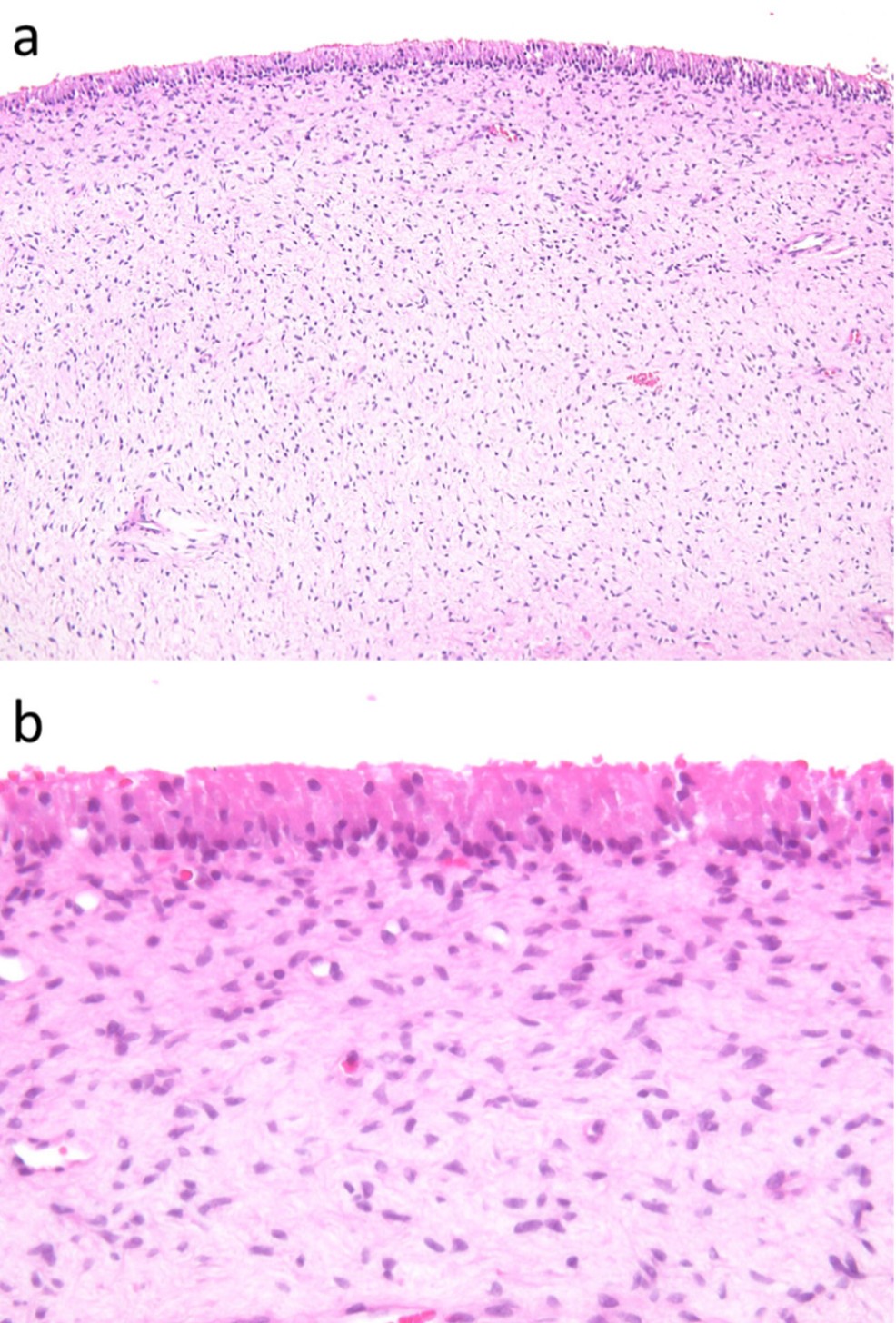

**Figure 3.** Histological findings of primordial odontogenic tumor. (**a**) The tumor is composed of cellular myxoid connective tissue (100×). (**b**) Tumor cells are spindle-shaped, lack cellular atypia and mitotic activity, and are covered by columnar epithelium (200×).

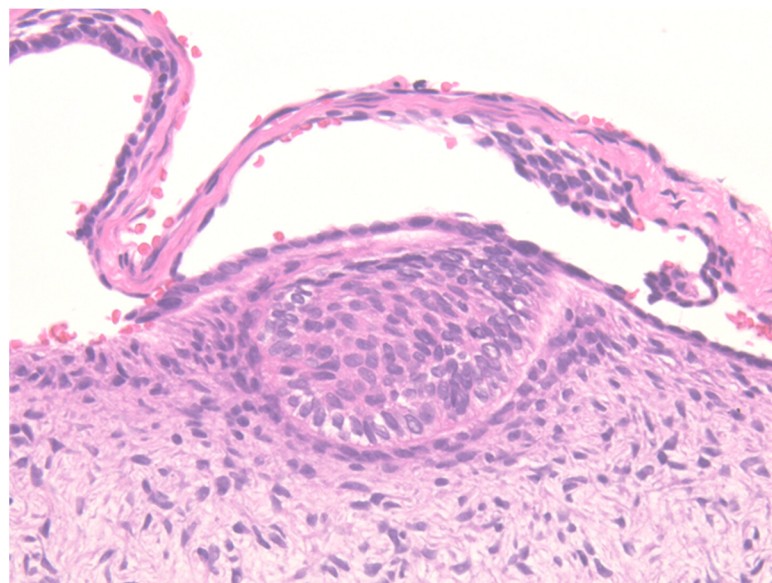

**Figure 4.** Histological findings of primordial odontogenic tumor. Nests of epithelium resemble dental lamina, and columnar epithelial cells show reverse nuclear polarization (200×).

Immunohistochemistry revealed vimentin expression in the epithelium and mesenchymal component, whereas syndecan-1 (CD138) and Bcl-2 were not expressed in the epithelium and mesenchymal component (Figure 5). According to histological and immunohistochemical findings, we diagnosed the patient as having POT. After the operation, the second molar erupted. Recurrence was not observed at the 11-month follow-up.

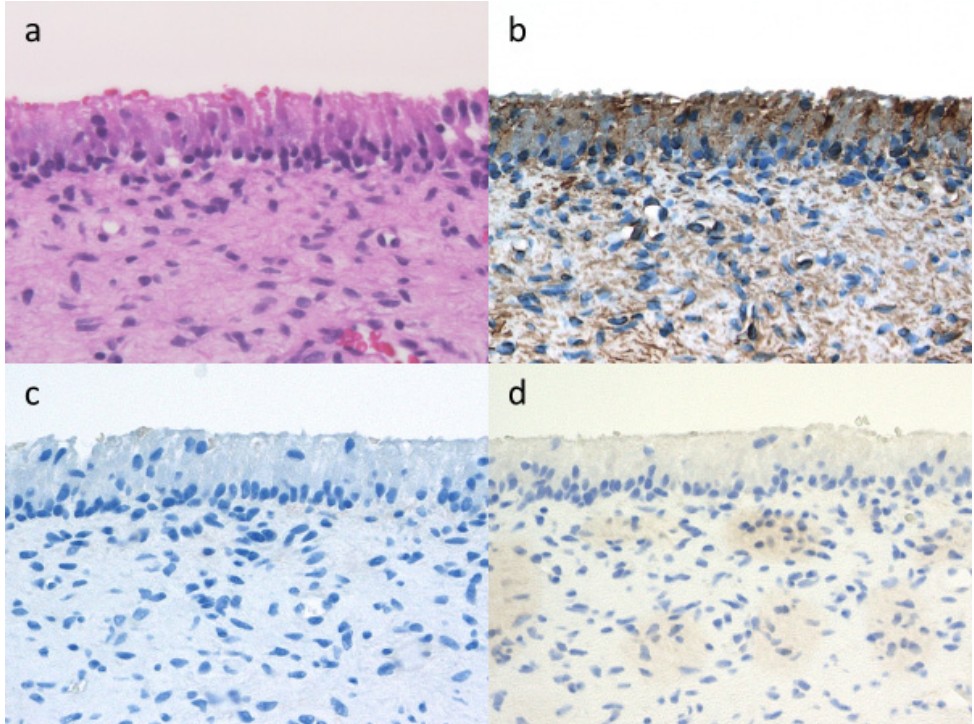

**Figure 5.** Immunohistochemical findings of primordial odontogenic tumor. (**a**) Hematoxylin and eosin staining (200×). (**b**) Vimentin expressed in the epithelium and mesenchymal cells (200×). Epithelium and mesenchymal cells did not express syndecan-1 (**c**) or Bcl-2 (**d**) (200×).

## 3. Discussion

POT is a new entity that was first reported in a case series of six cases in 2014. POT is rare tumor composed of cellular loose fibrous tissue surrounded by cuboidal to columnar epithelium.

The summary of clinical and radiological findings of POT cases in previous reports are described in Table 1. There was a slight male predominance (12 male, 9 female), with a mean age of 11.9 years (range: 2–19 years). Only 4 cases demonstrated maxillary involvement, whereas 17 cases exhibited mandibular involvement. All cases demonstrated well-defined radiolucent images in the jaws and tooth involvement. The size of the lesion can vary widely, from 8 mm to 90 mm. The clinical and radiographic differential diagnosis in a previous report indicated larger lesions measuring $17 \times 15$ mm to $90 \times 70$ mm were likely a benign odontogenic tumor, whereas smaller lesions measuring $10 \times 8$ mm to $35 \times 20$ mm were likely to be DC. Macroscopically, most cases presented a solid mass, while only one case demonstrated a cystic lesion. Finally, almost all cases exhibited no recurrences.

Moreover, the clinical and radiographic differential diagnosis was DC in 47.0% of cases (8/17). In our case, radiographic findings were a small well-defined lesion ($10 \times 7 \times 3.5$ mm) that circumscribed the tooth of the second molar. Therefore, the clinical diagnosis was DC.

However, the histological characteristics of POT are different from those from DC, and immunohistochemical findings can also help distinguish between POT and DC.

Histologically, POT is composed of variably cellular loose connective tissue, such as dental papilla, that is surrounded by cuboidal to columnar epithelium with reverse nuclear polarization. Conversely, DC comprises collagenous connective tissue and does not contain cellular loose connective tissue. Additionally, DC usually contains squamous epithelium without reverse nuclear polarization and does not have cuboidal to columnar epithelium.

There are three important differences in immunohistochemical markers between the present case and DC, which can help differentiate POT from DC. In DC and POT, respectively, vimentin, Syndecan and Bcl-2 were immunohistochemically examined in previous reports [3–10]. These three molecules were the common factor that were examined in both DC and POT. However, there is no report that compares the expression of these molecules in DC and POT in the English literature. Therefore, we summarized the immunohistochemical findings of three markers in Table 2. First, previous studies demonstrated vimentin expression in the epithelium in most cases of POT [3], which was also observed in this case. However, vimentin expression was not detected in DC [4]. Second, epithelial cells were shown to be primarily negative for syndecan-1 (CD138) in a previous report [5] and in the present case. However, in DC, almost all epithelial cells showed strong positivity for syndecan-1 (CD138) [6]. Syndecan-1 RNA accumulation is more intense when morphogenesis advances towards the cap stage, and it is also observed that syndecan-1 expression is lost during the bell stage. These facts led us to consider that this lesion may mimic the bell stages of tooth development. Finally, Bcl-2 expression was observed in some epithelial cells of POT [5], but was not detected in this case. Conversely, Bcl-2 positivity was 80% throughout the epithelial cells in DC [7]. Therefore, in the present case, the histological and immunohistochemical diagnosis was POT.

In conclusion, we reported a small well-defined lesion within the pericoronal region associated with an unerupted tooth that required distinction from DC. To our knowledge, this is the first report on the histological and immunohistochemical differences between POT and DC.

**Table 1.** Previous report of primordial odontogenic tumor (POT).

| Study | Age (Years) | Gender | Site | Radiographic Findings | Involved Teeth | Size (mm) | Clinical or Radiographic Diagmosis | Cut Surface | Recurrence, Follow-Up |
|---|---|---|---|---|---|---|---|---|---|
| Mosqueda Taylor et al. (2014) [2] | 3 | F | Mand | well-defined, biloculated | D, E, 6 | 90 × 70 | benign odontogenic tumor | solid mass | No, 9 years |
| | 3 | F | Max | well-defined | E, 6 | 35 × 30 | benign odontogenic tumor | solid mass | No, 6 months |
| | 13 | F | Mand | well-defined, biloculated | 6,7,8 | 80 × 50 | benign odontogenic tumor | solid mass | No, 3 years |
| | 16 | M | Mand | well-defined | 8 | 55 × 50 | benign odontogenic tumor | solid mass | lost of follow up |
| | 16 | M | Mand | well-defined | 8 | 65 × 50 | benign odontogenic tumor | solid mass | No, 10 years |
| | 18 | M | Mand | well-defined | 8 | 45 × 40 | benign odontogenic tumor | solid mass | No, 20 years |
| Slater LJ et al. (2016) [11] | 19 | M | Mand | well-defined | 8 | 25 × 19 | unknown | solid mass | No, 7 months |
| Ando et al. (2017) [8] | 8 | F | Max | well-defined | D | 16 × 15 | DC, benign odontogenic tumor | solid mass | No, 16 months |
| Mikami et al. (2017) [9] | 5 | M | Mand | well-defined | D, E | 80 × 80 | unknown | solid mass | No, 7 months |
| Bajpai and Pardhe (2018) [12] | 17 | M | Mand | well-defined, multilocular | 5, 6, 7, 8 | 30 × 20 | benign odontogenic tumor | unknown | No, 6 months |
| Almazyad A et al. (2018) [13] | 15 | F | Mand | well-defined, multilocular | 8 | 35 × 20 | DC | unknown | No, 3 months |
| | 18 | M | Mand | well-defined | 8 | 12 × 7 | DC | a yellow–tan mass | No, 20 months |
| Hatem Amer et al. (2018) [10] | 2 | M | Mand | well-defined, multilocular | impacted tooth | 30 × 40 | unknown | cystic lesion | No, 2 years |
| Bomfim B B et al. (2018) [14] | 4 | M | Mand | well-defined | D, E | 30 × 20 | unknown | solid mass | lost of follow up |
| Teixeira L N et al. (2019) [15] | 13 | F | Mand | well-defined | 8 | unknown | DC | unknown | No, 13 years |
| Poomsawat S et al. (2019) [16] | 17 | F | Mand | well-defined | 8 | 25 × 34 | other odontogenic cyst | unknown | No, 18 months |
| Wilson A. Delgado-Azañero et al. (2020) [17] | 12 | F | Mand | well-defined, unilocular | 5 | 30 × 25 | benign odontogenic tumor and DC | solid mass | No, 15 months |
| | 13 | F | Mand | well-defined, unilocular | 8 | unknown | DC | solid mass | No, 60 months |
| Naina S. et al. (2020) [3] | 14 | M | Max | well-defined, unilocular | impacted tooth | 30 × 20 | benign odontogenic tumor and DC | white nodule | No, 36 months |
| Kayamori K. et al. (2020) [18] | 10 | M | Max | well-defined, unilocular | D | 17 × 15 | benign odontogenic tumor | solid mass | No, 30 months |
| Present case | 14 | M | Mand | well-defined | 7 | 10 × 8 | DC | solid mass | No, 11 months |

Max: maxilla, Mand: mandible, DC: dentigerous cyst.

**Table 2.** Immunohistochemical findings of epithelium in this case and DC in previous reports [7,9,10].

| Antibody | POT | DC |
|---|---|---|
| vimentin | (+) | (−) |
| Syndecan-1(CD138) | (−) | (++) |
| Bcl-2 | (−) | (+) |

(+): positive, (++): strongly positive, (−): negative.

**Author Contributions:** S.O.: Pathology fellow responsible for working on the case, write up of the manuscript and final submission. H.K., S.S. K.T., and K.N.: Oral pathology assistant professor responsible for interpretation, review, and editing of final manuscript. H.N.: Oral pathology professor, consultant during the interpretation and getting to the final diagnosis. Responsible for review of final manuscript. T.Y.: Pathology professor, reviewed manuscript. All authors have read and agreed to the published version of the manuscript.

**Funding:** This research received no external funding.

**Institutional Review Board Statement:** Not applicable.

**Informed Consent Statement:** Informed consent was obtained from all subjects involved in the case report.

**Data Availability Statement:** Not applicable.

**Conflicts of Interest:** The authors declare no conflict of interest.

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
