# Peer review of "A Case Report of Primordial Odontogenic Tumor That Required Distinction from a Dentigerous Cyst"

_reports, doi:10.3390/reports4010004_

Round 1

Reviewer 1 Report

This study reported an extremely rare odontogenic tumor: Primordial odontogenic tumor that only 16 cases have been documented worldwide, and also only 2 cases have been reported in Japan. Meanwhile, this study is to focus on the exploration of differential diagnosis between POT and dentigerous cyst. The design of the study seems to have done well for the purpose that avoids being a cookie-cutter case report. However, there are some concerns about the flow of the manuscript and the description of the analysis.

Major points

  1. The importance of distinguishing the POT and DC needs to be described in the introduction in detail.
  2. While the POT is classified as a benign tumor, the tumorigenesis is still unclear. As a tumor, it still poses a proliferating potential as well as the possibility of cancerous changes. Therefore, it is necessary to emphasize the importance of distinguishing POT from DC in clinical practice.
  3. In figure 1, the representative radiographic CT images of POT is not reader-friendly and easy to understand. The ROI should be demarcated with either a white box or an arrowhead. For better clarity to showing the ROI, the stacks of CT images can be reconstructed into a 3D format.
  4. In the discussion,
  5. Line 82, the author mentions the three important markers which are vimentin, syndecan-1, Bcl-2, to distinguish the POT from DC. It should be described clearly of the reason of these three molecular became a candidate in this study for distinguishing the POT and DC. As far as we know that, there are several molecular expression has been detected in POT (ex. In epithelial tissue, amelogenin, CK19, and CK14, Glut-1, Galectin-3, Caveolin-1, Vimentin, p-53, PITX2, Bcl-2, Bax, and Survivin are positive; in mesenchymal tissue, Vimentin, CD90, p-53, PITX2, Bcl-2, Bax, and Survivin are positive.)

Minor point

  1. Keywords could be the Primordial odontogenic tumor; dentigerous cyst; odontogenic tumor;
  2. In Table 1. Legend, all change from Men to Male.

Reviewer 2 Report

Introduction 

Line 30-31. Please review the reference #2. Although this reference mentions the cases reported around the world, it has an editor letter that questions the case the authors report. (DOI: 10.1111/his.13999). 

Case report 

Line 52. Did they find any calcified areas and/or odontogenic epithelial islands or cords? Could it be possible to report this information? 

Will it be possible to describe if calcifications were found? 

Figure 2. Will it be possible to take a picture with of a slide with the surface of the lesion? 

Discussion 

Please redact this paragraph in order to make it more understandable and correct the style of the reference "POT is a new entity that was first reported in a case series of six cases in 2014. The authors described POT as a benign mixed odontogenic tumor1. To date, only two Japanese 60 cases have been reported in Japanv[4, 5]". 

Table 1, Please review the table 1 and also consider whether to include or not the manuscript made by Sun et al because they have an Editor letter in which questions the diagnosis made by them. (DOI: 10.1111/his.13999)  

I have some recommendations in their discussion section:

  1. Could it be possible that the authors discuss the immunohistochemistry relationship between tooth development (primordial tooth germ) and POT? 
  2. Could it be possible that the authors discuss the immunohistochemistry panel of POT? I recommend to review the reference (10.4317/medoral.21859) and, if possible, explain why they decided to use vimentin, Syndecan and Bcl-2 in your case.  

Table 1 

I recommend to review the manuscripts made by Delgado-Azañero et al that report 2 new cases of POT. (10.1016/j.oooo.2020.08.004), Naina S et al that report one case of POT (10.1177/1093526620972589) and Kayamori K et al that report one case with calcifications (10.1111/pin.13036). 

Please review if the name of 'Asma Almazyed et al' is well written.

Round 2

Reviewer 2 Report

Dear authors. The manuscript was improved in this new revision, now is better than previous version, and is more understandable. Now I recommend it for publication without changes.